# Assuring Energy Reporting Integrity: Government Policy's Past, Present, and Future Roles

Mohammed Hammam Mohammed Al-Madani [1] , Yudi Fernando [1,2] and Ming-Lang Tseng [3,4,*]

1   Faculty of Industrial Management, Universiti Malaysia Pahang, Gambang 26300, Pahang, Malaysia
2   Management Department, BINUS Online Learning, Bina Nusantara University, Jakarta 11530, Indonesia
3   Institute of Innovation and Circular Economy, Asia University, Taichung 41354, Taiwan
4   Department of Medical Research, China Medical University Hospital, China Medical University, Taichung 40402, Taiwan
*   Correspondence: tsengminglang@asia.edu.tw

**Abstract:** This study investigates government policy influence on energy reporting integrity in the past, present, and future. The study attempts to identify the dominating key themes in energy reporting and explore the function of government incentives and policies in influencing the integrity of energy consumption reports. A thorough literature review screening and theme identification were conducted through a systematic review. The data used in this study are mainly derived from English-language journals acquired from reputable academic databases such as Web of Science and Scopus. Social network analysis was used to examine the data retrieved with the VOSviewer software. The findings demonstrate that the key themes of government policy, energy reporting, energy management, and integrity are strongly focused in studies on energy policy, climate change, energy efficiency, renewable energy, life cycle assessment, carbon emissions, and sustainability. These topics included energy management, renewable energy, energy efficiency, emissions reporting, and energy transitions. The results suggest that there is little empirical support for how government policy promotes and validates the accuracy and integrity of energy reporting. The findings offer potential strategies for removing energy policy development, implementation, and reporting barriers. This study found that transparent disclosure of a company's energy consumption attracts new investment. The integrity and transparency of the energy report attest to a firm's commitment to working toward sustainable development goals. The study recommends that the government should align energy policies with clear guidelines about transparent energy disclosure and reform the existing sanctions and incentives to enforce the law.

**Keywords:** energy management; energy reporting; government policy; energy reporting integrity; energy audit; management commitment

## 1. Introduction

Energy is a vital contributor to all sectors of the economy in general, and especially to industry. Nevertheless, using energy for production typically produces undesired emissions that impact the natural ecosystem and the climate globally [1]. Both researchers and government leaders are encouraged to focus on social welfare and energy-related emission assessments and their possible contamination of the environment [2]. Energy consumption and GHGs have contributed to climate change and global warming [3]. Fossil fuels for power generation have resulted in massive carbon dioxide ($CO_2$) emissions, contributing considerably to global warming. As a result, attempts have been made to reduce $CO_2$ emissions through capture, storage, and usage among industries [4]. Firms have adopted low-carbon operations strategies such as energy management because of the relevance of carbon emission reduction for environmental management. Energy management and environmental friendliness principles have revolutionised the manufacturing industry's competitive landscape [5].

However, urban developments and economic growth have recorded an upward trend in energy use in developing countries. The unsuitable energy consumption has increased the demand for energy from conventional sources. Fossil fuels contribute to global warming, air pollution, and rain acidity. Due to awareness of the environmental impact, global countries have reached a consensus on controlling and managing energy-related carbon dioxide emissions, especially in energy-intensive sectors such as industrial and commercial. Since conventional energy sources cause ecological problems [6] and significantly contribute to climate change, clean energy is preferable for meeting the public's energy needs [7]. Energy management has also shifted from being only technology-based to incorporating multidisciplinary practices. Such practices involve top management commitment, energy knowledge, and energy auditing [8].

Government interventions through energy policy will guide and shape the integrity of energy consumption reporting and energy management practices within the industrial and commercial sectors. In addition, energy policy has been the leading actor in enhancing energy management and developing energy efficiency and reporting [9]. However, a lack of information, uncertainty, and less motivation within the industrial and commercial sectors has threatened energy management practices' integrity and the transparency of energy reporting. The government enforces reporting transparency through energy policy and encourages users to cooperate closely. In addition, government regulation drives the industry to carry out its social and environmental responsibilities [10].

Governmental subsidies for conventional fuels may reduce energy efficiency [11]. Without a suitable policy, energy subsidies may result in higher energy use and carbon dioxide emissions. Elgouacem [12] argues that energy policy is unsuccessful when subsidies lead to increased energy use and carbon dioxide emissions due to low energy costs. As a result, the transition to renewable energy will be delayed [13]. Moerenhout et al. [14] discovered that energy policy support in the form of subsidies reduces competition for renewable energy sources. Rentschler et al. [15] stated that energy subsidies disincentivise investment in alternative energy, impede innovation and energy efficiency, and increase fiscal responsibilities. Reforming the subsidies will, however, raise the production and consumption costs of externalities related to fossil fuels. The right distribution of incentives for energy management and reporting is, therefore, necessary for the effectiveness of energy policy. For the transition to alternative fuels and energy efficiency, energy policy must be aligned with incentives to achieve transparent energy reporting targets.

Previous studies have employed the day-watchman technique [16] to strike a balance between the public interest and regulations on energy policy. The day-watchman strategy serves as a compromise, balancing the state's and the market's obligations in preserving the public interest. The day-watchman approach includes defined goals such as the ability to create regulations and standards, provide authorisations and permissions, ensure the legality of public regulations, carry out monitoring and surveillance, and mitigate and penalise [17]. This strategy creates a framework for regulatory actions in which the regulator (day-watchman) creates the game's rules, informs market participants, and enacts incentives or penalties to enforce the rules [18].

Energy policy aims to promote energy efficiency and disclosure of energy consumption data. According to Kontokosta and Toll [19], the accuracy of energy reporting will give prospective investors, purchasers, financiers, and governmental organisations useful information about an organisation's sustainability performance and financial expenditures [20]. In addition, transparent energy reports will make it easier for market participants to consider the energy characteristics of manufacturing firms or buildings, particularly the projected energy expenses and carbon footprint, when making decisions regarding investments, acquisitions, and the financing of those assets [21]. The quality of knowledge is projected to gradually drive the markets for commercial, industrial, and residential buildings toward greater efficiency as, for example, building owners make investments in energy upgrades to compete for occupants and customers. Similarly, firm owners would attempt to obtain green credentials to meet the requirements of local and international policies.

Therefore, the integrity of energy reports is subject to local and international government policies [22].

Many countries, such as the United States and European countries, have established energy and carbon reduction schemes. To achieve sustainability, energy disclosure policies have been expanded, mandating businesses, buildings, and organisations of a certain size to disclose energy use [22]. This has led to the creation of new data streams that could shed light on energy trends and inspire data-driven initiatives to enhance energy efficiency. Energy data can be used by service providers, such as energy consultants and third-party managers, to target new clients and assess the accuracy of their reporting by comparing it to that of other service providers [19]. According to Palmer and Walls [23], energy reporting policy and incentives act as stimuli to increase awareness and lower transaction costs about energy reporting. This can encourage managers and decision-makers to participate more actively in the energy disclosure process.

The policy's success in driving energy efficiency and disclosure improvement relies on incentives for energy use reductions [24]. For instance, the industrial sector can receive incentives to comply with government policies, such as income tax allowances [10]. In addition, companies need to comply with regulations and report their energy efficiency achievements as their social and environmental responsibilities. Transparency about sustainability disclosures has become indispensable as companies are urged to comply with triple bottom lines, primarily environmental issues. Therefore, energy reporting and efficiency improvement are strongly related to preserving the environment. In addition, mandatory sustainability reporting and incentives for successful firms will motivate other firms to adhere to regulations and policies [25]. Moreover, this reporting will help the government to tackle issues related to environmental concerns.

Our study offers insights into current government policy trends toward the integrity of energy reporting while also discussing gaps in the literature. The contribution of this study is conceptualising a framework that allows scholars to connect and further extend the driven behavioural theory of energy reporting integrity. The research contribution is the extensive literature review on energy reporting and government policy and determining the gaps. Limited studies have been carried out on this topic, and further studies are still needed to enhance energy policy effectiveness. This study also has implications for the trends and gaps in the energy management literature. To accomplish this goal, the authors attempt to use bibliometric analysis for policy and energy reporting papers. The findings demonstrate patterns, gaps, and a visual image of the evolution of this field of study. This approach helps to review and reveal the main themes of government policy studies as the enabler for the integrity of energy reporting and the effectiveness of energy management.

Providing transparent reports on energy use necessitates significant effort from all parties involved, including legislators, building owners, business stakeholders, and investors. Before anyone can act based on an energy consumption report, it must first be collected, validated, processed, analysed, audited, and disclosed. To design new regulations for information disclosure, policymakers, building owners, business stakeholders, experts, and academics must find an optimal equilibrium between the costs and integrity of reports. Subsequently, critical considerations arise given the considerable variation in the literature and the legislation that legal systems and scholars have passed. Therefore, this study intends to answer the following questions to understand the rising domain of government policy toward energy report integrity:

- What are the key themes in energy reporting integrity? This research question will help to clarify both the dominant topics and the gaps in this research field.
- What is the role of government policy in energy reporting integrity? The response to this question provides an overview and an interpretation of the essential publications in the public policy field, making it easier for researchers interested in government policy studies to understand the latest developments.
- What are the highest co-occurrence keywords of energy reporting in the literature? This point makes it possible to understand the most used words in studies on energy

reporting integrity and the field's main components that apply to researchers, to facilitate progress in the research. They must acquire a comprehensive framework.

- What will be the future research of energy reporting integrity studies? The answer to this question can be used to create a research strategy that will aid in developing the topic.

Previous research has looked into government policy impacts on the energy efficiency gap [26], mandatory reporting on corporate social responsibility [27], energy policy evaluation during the climate crisis [28], stakeholder views on the greenhouse and energy reporting [29], voluntary GHGs reporting in a market governance system [30], energy reporting in energy-intensive industries in Malaysia [31], and the waste to energy supply chain [32]. However, increased energy consumption is a global issue, and most firms' annual reports tend to be declarative.

On the contrary, this study focuses on government policy and energy reports' integrity. This paper explores how previous studies addressed government policy support in shaping the integrity of energy usage reporting. We proposed a conceptual framework based on a thorough review of energy management literature. The framework is conceptualised based on literature analysis on energy disclosure integrity and government policy to better understand energy reporting integrity's future direction. The paper uses a systematic review technique to identify the gaps and key themes in energy reporting integrity and government policy. Energy reports' integrity has not been thoroughly investigated as a standalone subject but as a sub-topic or is merely implied in the best cases.

The current study consists of the following sections. The study background is discussed in the first section. The following section is the literature review. The next part is the methods section, covering the approach used to obtain the research findings. The results are then presented and critically evaluated in the results and discussion section. Lastly, the discussion section will cover the findings, research implications and conclusions, limitations, and future recommendations.

## 2. Literature Review

Reducing the world's energy consumption to a manageable level is a critical component of fulfilling the commitments of the Paris Agreement and carbon neutrality [33]. A shift in energy end user behaviour is required because of the intensity and scope of the climate issues and the ongoing worldwide increase in energy consumption [34]. Apart from that, expenditures on energy-saving technologies are indispensable to managing energy consumption [35]. Energy policies have focused on supporting energy-efficient practices such as financial incentives for innovative technologies [36]. However, energy savings have been lower than expected in numerous cases due to a rebound effect or exogenous factors such as population or economic expansion. The conventional energy efficiency programs are still important, but they are insufficient to meet the Paris Agreement objective and reduce energy demand quickly.

The main objectives of energy policies are to achieve sustainable development goals in terms of sustainability, resource use efficiency, environment conservation, and ensuring high-quality services to stakeholders. Many government initiatives are presented in policies and plans [37]. Governments determine the suitable energy policy for energy production, distribution, and consumption. Gasoline, coal, and natural gas are the significant targets of carbon taxes in some jurisdictions and countries. Furthermore, policy and regulation have played an essential role in reducing $CO_2$ emissions [38]. The carbon tax is one of these measures, a price the government sets for emitters to pay for each tonne of $CO_2$ released. The carbon tax might be based on the amount of $CO_2$ emitted by a corporation or a tax placed on $CO_2$-intensive products or services [4].

Based on Opschoor's classification of environmental policies, Ayala et al. [39] proposed categorising energy efficiency programs into three major categories: (i) command and control (for example, building codes and appliance standards); (ii) pricing strategy (for example, taxes, subsidies, tax deductions, credits, permits, and tradable responsi-

bilities); and (iii) data mechanisms (labels, audits, intelligent meters, and information). Shen et al. [40] divided policy tools into three categories: (i) mandatory administration instruments, (ii) economic incentive instruments, and (iii) voluntary scheme instruments. Shen et al. [40] divided the three elements into three sub-categories as well: (i) legislation and regulation, including codes and standards; (ii) subsidies, tax, and loan incentives; and (iii) research and development, certification, and labelling, as well as government services. Sterner and Robinson [41] identified four types of policies: (i) price-type regulation (taxes, subsidies, and fees); (ii) rights with quotas (tradable permits, property rights, and certificates); (iii) quantity-type regulation (efficiency standards and restrictions); and (iv) informational/legal regulation (information reporting and voluntary agreements.).

Previous research on climate change and corporate social responsibility has evaluated companies' disclosures due to social pressure. Some studies have covered energy-related information, although only a few firms have disclosed energy-related information [42]. Despite that, sustainability reports' energy consumption reporting seems to be minimal. However, governments can commence environmental management programs and increase energy efficiency among end users. It is easier for governments to benchmark energy consumption when energy use disclosure is available. Energy consumption reports should be made mandatory to benchmark in energy-efficient industries. Government and industry can also arrange the industry energy intensity ratio. Energy management and consumption can be effectively managed with widespread stakeholder participation. Government can push a national energy policy for the corporate sector with the help of these industry-backed initiatives. According to Nurgazina et al. [43], national-level strategies should consider sectoral needs due to varying levels and energy consumption patterns across energy-intensive to non-energy-intensive industry sectors.

Sustainability reports encompass various topics, but the environment, particularly emissions reports, is the key focus [44]. Therefore, insights about the government policy determinants in energy reporting integrity are critical, especially when considering the current tendencies toward emphasising various stakeholders in the monitoring, measuring, and documenting of GHG emissions and energy consumption. Sustainability reports have become much more popular worldwide, and these reports are also increasingly more extensive [45]. According to Balogh et al. [46], 95% of the Fortune Global 500's most significant 250 corporations now report on their sustainability actions. The most effective reporting rates are observed in European countries. While reporting rates in the United States are lower than in Europe, the trend in the United States is toward higher reporting.

Adopting policies, design principles, and technological advancements that raise the proportion of energy-efficient goods and services in a given market constitutes one of the most promising public policy tools for accelerating market transformation around energy efficiency and energy disclosure [47]. In addition, firms will start benchmarking with successful businesses adopting energy efficiency practices and disclosing energy use. For instance, Local Law 84 in New York City mandates that all facilities of 50,000 square feet or larger must disclose their energy consumption annually. Six cities have approved disclosure laws since New York City adopted the ordinance, and more than 20 other cities are considering doing the same [48].

Managers may opt not to share negative energy use data to preserve their company's profitability [49]. However, not sharing such data could elevate the commercial risk for a firm. Nevertheless, disclosure of unfavourable information about environmental liabilities (such as pending legal actions and fines) that are not disclosed in a company's financial statements has a detrimental effect on the value of the company, and the disclosure of unfavourable information enhances the credibility of the information that is disclosed [50]. A company develops a solid reputation for offering excellent disclosure of energy use. Stakeholders agree that companies are more responsible if they publish higher-quality information.

However, legitimacy and stakeholder theories may also offer a more comprehensive understanding of the integrity of energy reporting since they acknowledge that political

and social frameworks define enterprises in addition to institutional frameworks [51]. For example, according to [52], the theory behind social and energy reporting is based on the premise that disclosing energy information to stakeholders is a more effective way to justify a company's continued existence. Therefore, transparent and audited energy disclosure can be seen as a symbolic means by which a company can inform the public to regulate its political or commercial position [53].

The International Auditing and Assurance Standards Board (IAASB) created ISAE 3410—Assurance Engagements on Greenhouse Gas Statements—in response to the importance of disclosure of firms' GHG emissions [54]. While the number of sustainability reports that receive third-party assurance is growing [55], other reports do not have external assurance [54]. While a prior empirical study on environmental audits saw environmental audits as a "universal sort of management practice," Ding et al. [56] imply that firms can undertake internal or external audits. According to Ding et al. [56], internal audits are standard in environmental reporting, with over 60% of the companies in their survey indicating that they have created internal environmental audits.

To the best of the authors' knowledge, while writing this study, limited evidence and studies focus on how government policy helps shape energy reporting integrity.

## 2.1. The Day-Watchman Approach for Energy Policy and Incentives

The day-watchman strategy aids in addressing policy deficiencies while safeguarding society and the general public's interests [11]. The degree to which state institutions act, explicitly or implicitly, in addressing energy concerns, such as transparent energy use disclosure through legislation, can be used to examine the challenges for policy intervention through subsidies and government incentives reform [57]. For instance, a government's yearly budget must be approved by parliament each year, in which incentives or subsidies are essential components. The day-watchman strategy can be helpful in this situation. Better policies are developed by the regulator (government), who provides information to the energy users about transparent disclosure and uses punishments to enforce the law [18]. The alignment of policy intervention through incentives helps to improve energy use and reporting efficacy and assists the government in meeting its obligations toward emissions reductions.

## 2.2. Energy Reporting Integrity as Market Driving Force

Policies requiring energy disclosure have a significant chance of changing consumer perceptions of energy efficiency. Similar disclosure regulations, such as those governing the auto fuel economy and nutrition information on menu items offered by restaurant chains, have been demonstrated to alter the behaviour among suppliers, manufacturers, consumers, and end users [19]. For instance, more information on energy performance in the building sector would enable renters (consumers) to integrate energy measures into leasing considerations. Consequently, there should be greater demand for energy-efficient buildings, enhancing asset value and motivating building owners to increase the relative energy efficiency of their premises [58].

In a market-based economy, inefficient resource allocation is hampered by information and incentive issues [59]. The accessibility and expense of financing for energy efficiency measures is another issue. In addition, the funding sources are more limited due to the uncertainty and risk surrounding the anticipated energy savings [60]. These uncertainties, mainly due to a lack of timely, accurate, and transparent data, make it possible for capital to be allocated to investments in energy efficiency below what might be considered a societal ideal. According to Papadopoulos and Kontokosta [61], energy consumption reports gathered through energy disclosure laws can increase the quantity and quality of information on energy efficiency and aid in removing doubts regarding energy consumption, the savings anticipated from an energy retrofit, and energy conservation measures.

The integrity of energy reports has a big impact on private investment, especially regarding the power network [62]. Municipalities and energy providers can more effectively

decide where to focus their capital funds and how to best target energy efficiency incentives by identifying clusters of inefficient or energy-intensive firms or buildings [63]. The finding of possible sites for decentralised generating stations and other shared renewable energy capacity types could also be accomplished using transparent energy disclosure [50]. The ability to properly manage growth prospects and handle the hidden costs of ineffective facilities, such as pollution levels, greenhouse gas emissions, and other societal problems, is a benefit of comprehending the geographical trends of energy demand [64].

Energy performance data can support and accelerate market change in the business sector. One issue with expanding energy conservation efforts has been the result of the efficiency market being extremely small, or the range of facility energy efficiency measures and exchanges or conversion of environmentally harmful facilities being limited [65]. By fostering competition in the market around energy efficiency and, potentially, wastewater, pollution, and other resource inefficiencies, the dependability of energy consumption information may cause alterations in market behaviour [66]. Nevertheless, it is initially required to define the process of information flows and its application in policy with multiple parties that affect energy efficiency better able to comprehend the mechanisms of market transformation and shifting demand [67].

For a positive effect on market behaviour, energy consumption statistics and energy disclosure regulations must be collected, evaluated, and disseminated to assist the user in making decisions and benefit both the providers and consumers of this data [68]. The utility company is where energy usage data is first created before being given to facilities managers. Service charges or more sophisticated real-time data access can be used to acquire information on energy use. In return, the operator must submit consumption information to the relevant government agency if the regulation covers the facility. Enterprises in markets with higher external finance needs have higher levels of reporting, providing information that gives stakeholders equitable rights to a company's financial and other information [69]. Prior research has shown that a firm's disclosure reduces data asymmetries [70]. Information asymmetry increases investment uncertainty, thus lowering the expected returns [71]. It is conceivable that a corporation incurs expenses because of a more open policy of disclosure of environmental information. The authors of [72] have found that firms do not disclose all relevant information because doing so could potentially impact future cash flows.

### 2.3. Integrity Concept in Energy Reporting

Integrity means having the following moral qualities: soundness, honesty, being corruption-free, and particularly ensuring contracts' trust-worthiness [73]. In this study, integrity in reporting energy consumption implies the soundness of reports, the trustworthiness of data, accurate reports, and transparent reports. Public opinion, institutional entities, socially aware stockholders, multinational corporations, and national and international legislation are all pushing for more openness in the non-financial reporting [74]. Firms are frequently scrutinised for their influence on ecosystems and territories [75]. Stakeholders demand that information about the environment, governance, and energy be disclosed transparently.

Integrity in non-financial reporting, such as energy consumption reports, refers to transparent reports that organisations publish to provide information on non-financial factors not considered in traditional financial statements. Non-financial reporting methods, as a result, provide stakeholders with a much more complete image of organisations than standard financial reports. Nevertheless, they are inextricably linked to organisations' operations and their effects on the environment.

International Financial Reporting Standards (IFRS) are a well-known global accounting standard founded on valid and verifiable financial information supplied to investors. They are used in 144 different jurisdictions around the world. On the other hand, sustainability reporting falls under ESG (environmental, social, and governance) reporting. The factor that makes companies reluctant to produce an energy consumption report is that they are

not transparent in running their business. The second factor is that the companies consider the energy consumption report unnecessary. Moreover, no regulation requires a company to release an energy consumption report expressly, or only voluntarily in the best cases (Kim, 2019). Therefore, there should be efforts from governments to encourage transparent sustainability reports in general and energy consumption reporting in specific [76].

Most previous research on sustainability assurance has focused on the content of these assurance statements; for example, Xiao and Shailer [77] argued that the reliability of sustainability reports increases stakeholder confidence in the information's quality. García Alcaraz et al. [78] investigated the voluntary market for sustainability assurance and discovered a strong correlation between requirements for increased integrity and a report's reliability. Similar work has used interviews to learn more about how companies try to legitimise integrity with essential audiences and how practitioners establish the profession of sustainability integrity [79]. While the current study focuses on energy report integrity, [80] states that internal audits play an essential role in sustainability reporting, citing survey data from over 2000 companies that publish environmental reports showing that over 60% used internal audits for integrity assurance.

### 2.4. Energy Management Practices

The focus of energy management in the late twentieth century was solely on technological advancement. Since 2000, however, the emphasis has switched to multidisciplinary techniques, including efficient behaviour, housekeeping activities, and energy management practices [81]. Despite the numerous potential benefits of energy measures, their adoption has been restricted. The term prolonged energy efficiency gap has been coined to describe this situation [82]. Several research papers and academic and business entities have studied the energy management process. In the last several years, this subject has experienced considerable advancements, the most notable of which is the introduction of the IS0 50001 Energy Management Standard [83]. The standard applies to all energy users; it is designed to help significant consumers improve the quality of their energy management systems. Addressing environmental concerns depends on the energy-intensive sectors' participation [84].

A previous study has shown that energy knowledge is critical to energy efficiency [82]. Because an energy auditing committee must review the data, management and reporting should be open, and energy usage records will be audited to see any potential for energy savings. As a result, there is a chance of lower energy use, lower GHG emissions, and more transparent sustainability reporting [85]. At the same time, top management commitment is required to assist and instruct energy teams in identifying energy inefficiencies in processes, machines, and buildings. The management commitment should improve energy awareness to increase engagement in developing the energy-saving options identified.

### 2.5. GHGs Emissions Reporting

Climate change is a worldwide issue, and the United Nations Framework Convention on Climate Change (UNFCCC) establishes an overarching framework for global efforts to address the concern. The Kyoto Protocol is a global consensus connected to the United Nations Framework Convention on Climate Change (UNFCCC) that establishes the objectives for industrialised nations to reduce greenhouse gas (GHG) emissions [86]. Organisations will gain directly by monitoring and reporting GHG emissions since their energy and resource costs will be reduced. Another advantage is that they will have a better awareness of their exposure to climate change risks and will be able to demonstrate leadership, which will help them build their green credentials in an increasingly environmentally sensitive market [87]. Several organisations are requesting information on greenhouse gas emissions from their suppliers, and many small firms will be expected to measure and report their emissions in the future [76].

Several mandatory or voluntary public programs have evolved since the late 1990s, encouraging or empowering businesses to measure and report their GHG emissions. These

requirements are part of environmental and non-financial disclosure requirements, and policy instruments set a carbon price, such as carbon taxes, emission trading programs, and stock exchange listing requirements [68]. Due to the growing number of reporting schemes, the number of organisations or entities reporting under required or voluntary schemes has increased [88]. Recent trends show that the number of government schemes is increasing, with some countries demonstrating a variety of schemes in operation or development at both the subnational and national levels. The EU ETS, for instance, includes $CO_2$ emissions from nearly 11,000 facilities in 30 nations. In 2009, over 11,000 businesses in Japan reported their $CO_2$ emissions under the mandatory GHG Accounting and Reporting System, accounting for nearly half of the country's total emissions. By contrast, in 2010, around 6700 businesses in the United States reported data under the GHG Reporting Program, accounting for nearly 80% of total GHG emissions [49].

Governments confront several obstacles to implementing GHG emission reporting methods in underdeveloped nations. The primary motivation for governments to demand GHG emission data from businesses is to push businesses to reduce their GHG emissions while making this data available to investors. Furthermore, governments use the data for various purposes, including supporting existing emission trading programs, supplementing domestic climate change policies, and revising national GHG inventories [89]. Most government GHG reporting programs (especially those linked to emission trading schemes) require enterprises to release their GHG emissions publicly. On the other hand, some institutions go far further, requiring firms to report on their carbon reduction goals and other climate-related information. Furthermore, providing the requisite policy coherence and coordination of various pieces of legislation (e.g., combining carbon reporting with other reporting requests) and putting in place the proper incentives to drive firms to cut emissions are essential [90].

## 3. Methods

Using a bibliometric technique and analysis proposed by Shabir et al. [91] and Wang et al. [92], this study investigates the research questions indicated in the introductory section. VOSviewer version 1.6.8 was used to conduct the bibliometric analysis in this work and can be used to identify literature trends and gaps in databases [93]. The method used in this study works well for bibliometric clustering. Furthermore, the software is designed to download papers from databases such as Web of Science and Scopus and upload the files directly into the software. Furthermore, for computer-literate researchers, interpreting the results is simple.

This study increases the dependability of previous studies' conclusions by critically reviewing and increasing the volume of papers about government incentives, energy policy, energy management, and disclosure integrity. We removed books from this investigation using a keyword filter for journal article searches. We included a science discipline/perspective to the study to further the theoretical understanding of energy report integrity and to broaden the study's scope beyond just the sustainability and environmental management disciplines. Additionally, we opted not to restrict the study by publication year out of fear that it could compromise the content analysis. In the subsections that follow, we go into further detail on the steps required in operationalising the stages and phases of our methods, including an overview of content analysis, social network analysis, and the research process.

### 3.1. Content Analysis

Keyword occurrence in a text serves as the foundation for content analysis. Analysing the frequency of keywords used in literature allows for determining the key themes of the literature [94]. Keywords are an important part of content analysis. Thus, the strategy used in this study was to use the keywords integrity of energy reporting, government incentives, and energy management to categorise articles in databases, while filtering out other undesired outcomes yields the most accurate and desirable findings. Leading databases'

keyword selections for data analysis provided a positive signal of whether the chosen databases were appropriate for energy studies. Additionally, by using bibliographic data, authors could determine which database offers greater search accuracy and contributions to energy literature.

For a multidisciplinary investigation, keyword frequency or bibliographic analysis alone may not be enough, according to Wang et al. [92]. When examining the integrity of energy reports, environmental management, energy management, and energy policy are complicated. For instance, energy consumption reporting and energy efficiency are linked to energy policy, whereas carbon emissions and climate change are connected to environmental management issues. Wang et al. [92] advise investigating the association between the co-occurrence of a keyword to find keyword clusters. After evaluating the content analysis by keyword occurrence, social network analysis was employed as an additional tool to analyse, interpret, and identify thematic information about the subject under investigation. Before going into greater detail about the stages and steps involved in this study's entire content analysis approach, we first highlight the significance of social network analysis using VOSviewer software.

*3.2. Social Network Analysis*

The method used in this study is ideal for developing conceptual models and bibliometric grouping. Furthermore, research can identify patterns and gaps in the published literature and databases using the software tools employed in this study, which are freely available. The benefit of clustering publications utilising open-access tools such as Mendeley and VOSviewer is that it does not require highly developed computer literacy or an extensive understanding of clustering algorithms. For instance, provided there is no data duplication, data downloads from journal databases can be directly uploaded to the software tools without the requirement for preprocessing. Since Mendeley is the perfect tool for avoiding data duplication, it was utilised to filter the data for this study.

Although both Mendeley and VOSviewer are simple to use, it is nevertheless advisable to have a basic understanding of clustering techniques to perform consequential analysis and help interpret the acquired results. Using a co-occurrence keyword, social network analysis (SNA) discovers word clusters [94]. In this way, SNA makes it possible for the techniques employed in this study to pinpoint the most popular journals and energy reporting themes. Utilising VOSviewer version 1.6.8 software, we enabled SNA for this study [94].

Utilising the co-occurrence-based keywords method, VOSviewer can perform word frequency analysis and is used to execute the bibliographic analysis. The program uses mappings, aggregating, and normalising procedures and offers the potential for data visualisation. This software uses the Apache OpenNLP toolkit to perform part-of-speech tagging (i.e., to identify verbs, nouns, and adjectives). A linguistic filter is then applied to isolate noun phrases. The filter chooses all word sequences that only contain nouns and adjectives that finish in a noun. Therefore, VOSviewer determines a relevance score and terms pertinent to the topic of interest for each noun phrase. The visualizing capability of the software shows clusters as circles with varying sizes of circles that show the concentration of publications in a particular category. The more tightly two topics are related, the closer they are located tied to one another, and frequently quoting the same sources. Different colours denote groups of publications with stronger connections [91].

*3.3. Research Process*

This study builds on earlier research on energy management, energy policy, government incentives, and energy reporting [95–97]. The content analysis process utilised in this paper is summarised in Figure 1. First, we separate this process into several stages. Stage 1 required researching to acquire all articles focused on energy reporting, regardless of the study field. Stage 1 was the selection of literature from the Web of Science and Scopus databases. A co-occurrence analysis was conducted that used the acquired

literature to identify current research themes. Once a thorough search was conducted of the data selection in stage 1, a filter was applied for the keyword to include government policy, energy reporting, integrity, and energy management and excluded book chapters. Duplicate removal was used in Stage 2 to reduce literature redundancies. The total number of publications after the filtering was 259. Moving on to Stage 2, after using Mendeley auto-check for duplicates, a thorough examination for duplicate papers was performed to account for database filter limitations. Then, once stage two was completed, Stage 3 covered uploading the data into VOSviewer software to conduct the analysis. Stage 3 assisted the authors in identifying literature gaps and prospective study areas of interest. Lastly, In Stage 4, the authors analysed the available results obtained from VOSviewer to find thematic links and build reflections for the research. Stage 4 entails interpreting the outcomes of the Stage 3 analysis. The four stages must be followed by content analysis to be reliable. Therefore, content analysis for the 259 articles was conducted by analysing the literature content on government policy, energy management, and the integrity of energy reports and categorising them into themes.

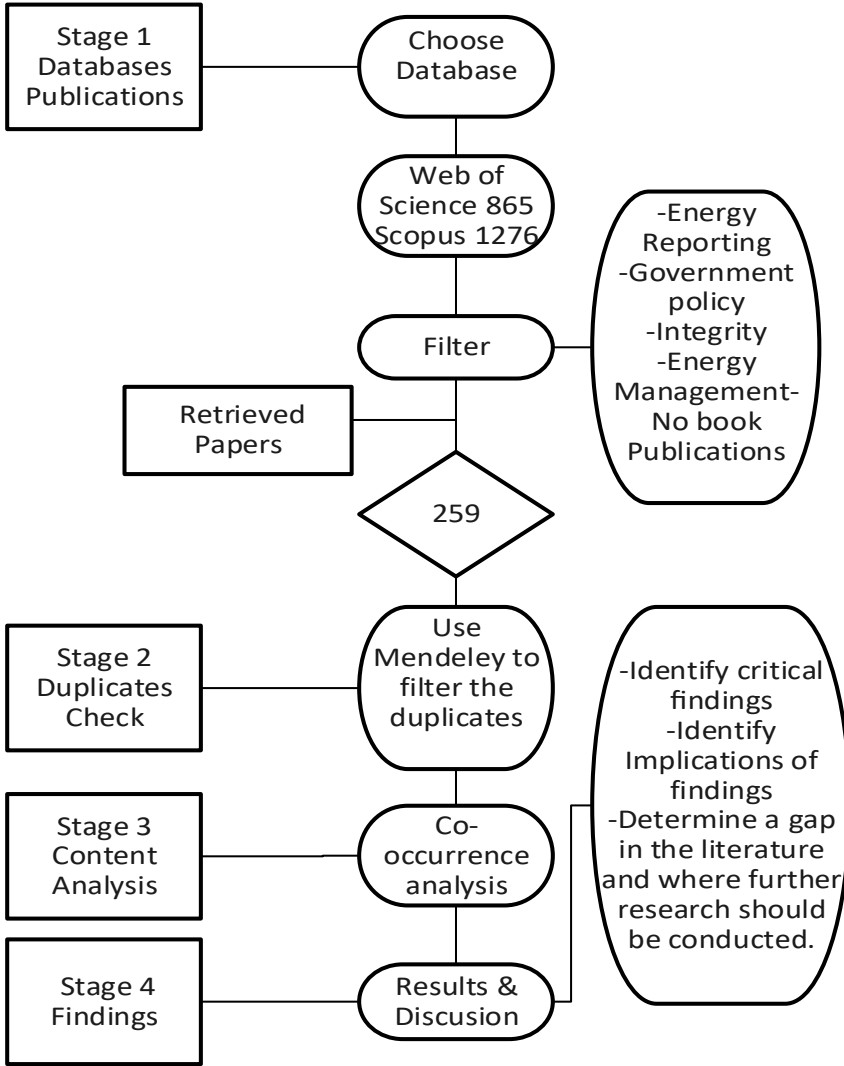

**Figure 1.** Research process.

## 4. Results

This section may be divided into subheadings. It should provide a concise and precise description of the experimental results, their interpretation, and the experimental conclusions that can be drawn.

The findings obtained from VOSviewer show its effectiveness as a tool for publications clustering, especially when grouping the content and themes of publications. The utilisation of VOSviewer provides helpful infographics and an easily understood visualisation of the clustering content. The software can also identify keyword occurrences density effectively at an aggregate level. Moreover, VOSviewer is very helpful when determining gaps in analysis or looking for specific patterns and topics among so much data. The visualisation function can determine the occurrence density of keywords in publications, even at an aggregate level, through the VOSviewer software. The visualisation function was beneficial for analysing the clusters obtained after the extracted data was run. VOSviewer software can show research gaps and changes in research trends for a particular topic or field of study. This software provides a good guide for future researchers interested in topics who want to conduct dynamic research.

The key themes of government policy, energy reporting, energy management, and integrity are extensively concentrated in studies related to energy policy, climate change, energy efficiency, renewable energy, life cycle assessment, carbon emissions, and sustainability. The five clusters identified by co-occurrence analysis are depicted in Figure 2. These findings aid in identifying the major themes relating to government policy on energy management and reporting. These findings show that sustainability is the most dominant theme (cluster 1). Thus, sustainability is the central energy management and government policy cluster and the most widely debated keyword. In addition, sustainability studies are linked to environmental management. The other clusters are related to climate change (cluster 2), circular economy (cluster 3), energy policy (cluster 4), and governance (cluster 5). Each cluster has a sub-category. For example, the cluster 1 theme is sustainability, covering the following concepts: carbon footprint, corporate social sustainability, economy, climate change, energy policy, governance, renewable energy, life cycle assessments, and ecosystem services. Figure 2 shows the sustainability cluster and the concepts it links to sustainability.

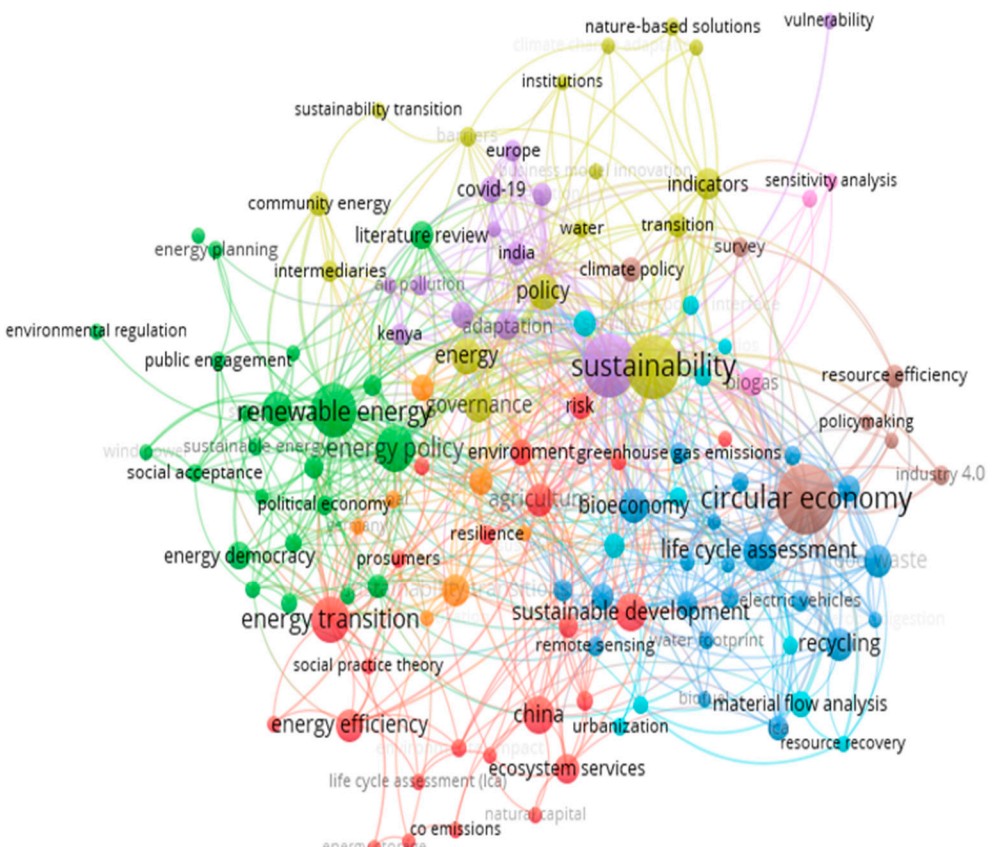

**Figure 2.** Network visualisation bibliographic coupling keyword occurrences.

Cluster 2's theme is climate change, covering the concepts of climate policy, energy policy, renewable energy, environmental impacts, sustainable developments, bioeconomy, stakeholders' engagement, and lifecycle assessments. Figure 2 shows the links established concerning climate change in literature. Cluster 3's theme is the circular economy. The theme of circular economy covers concepts such as innovation, resource efficiency, environmental impacts, sustainable development, energy policy, renewable energy, recycling, material flow analysis, and lifecycle assessments. Cluster 4's theme is energy policy. The energy policy theme covers the following concepts: energy democracy, energy planning, sustainability transitions, energy efficiency, and climate policy. Lastly, Cluster 5 is a governance theme. The governance theme covers topics related to the Paris Agreement and its adaptations, sustainability, renewable energy, energy policy, community energy, and prosumers.

The findings revealed useful information regarding the keyword energy report integrity and distribution of government policies, laying a solid foundation for keyword co-occurrence frequency, as shown in Table 1. Keywords were extracted from articles using VOS viewer's text-mining tool. This function generates a two-dimensional map with a co-occurrence network of keywords (adjectives and nouns). When two keywords appear in the same title/abstract, they are said to co-occur, and keywords with a higher co-occurrence rate are found closer together. VOSviewer software was utilised to create the co-occurrence map in this study. The number of clusters was decided based on interpretability factors, and binary counting was used. A keyword had to appear at least eight times.

**Table 1.** Keyword co-occurrences.

| Keyword | Weights | Link Strength |
|---|---|---|
| Biodiversity | 9 | 6 |
| Bioeconomy | 20 | 17 |
| Bioenergy | 11 | 8 |
| Biogas | 13 | 20 |
| Carbon Footprint | 10 | 13 |
| Circular Economy | 79 | 83 |
| Climate Change | 66 | 75 |
| Climate Policy | 10 | 6 |
| Community Energy | 10 | 14 |
| Ecosystem Services | 15 | 7 |
| Electric Vehicles | 8 | 4 |
| Energy | 21 | 32 |
| Energy Democracy | 13 | 15 |
| Energy Efficiency | 18 | 7 |
| Energy Justice | 20 | 26 |
| Energy Policy | 34 | 31 |
| Energy Security | 11 | 10 |
| Energy Transition | 34 | 21 |
| Environment | 11 | 15 |
| Environmental Sustainability | 8 | 9 |
| Governance | 19 | 20 |
| Greenhouse Gas Emissions | 8 | 9 |
| Life Cycle Assessment | 25 | 29 |
| Machine Learning | 9 | 10 |
| Material Flow Analysis | 12 | 11 |
| Mitigation | 9 | 16 |
| Policy | 22 | 25 |
| Policy Analysis | 8 | 18 |
| Recycling | 18 | 19 |
| Renewable Energy | 46 | 50 |

**Table 1.** *Cont.*

| Keyword | Weights | Link Strength |
|---|---|---|
| Resource Efficiency | 9 | 11 |
| Sustainability | 67 | 81 |
| Sustainability Transitions | 18 | 17 |
| Sustainable Development | 24 | 13 |
| Sustainable Development Goals | 10 | 10 |
| Waste Management | 12 | 22 |

Comparison of countries regarding scientific performance is noteworthy, particularly in developing disciplines such as energy management, energy policy, governmental incentives, and energy disclosure. The publication distribution by country for the 259 publications used in this study is shown in Figure 3. This analysis places a lot of emphasis on the number of publications as an indication of research performance by country in the energy reporting field. The findings demonstrate that China, the United States (US), and England are the top three national producers of energy publications. It is not surprising that China has the largest number of publications because of its industrial expansion and larger population. The publication count indicates that nations and governments have raised awareness about the importance of energy disclosure and the sustainable use of resources.

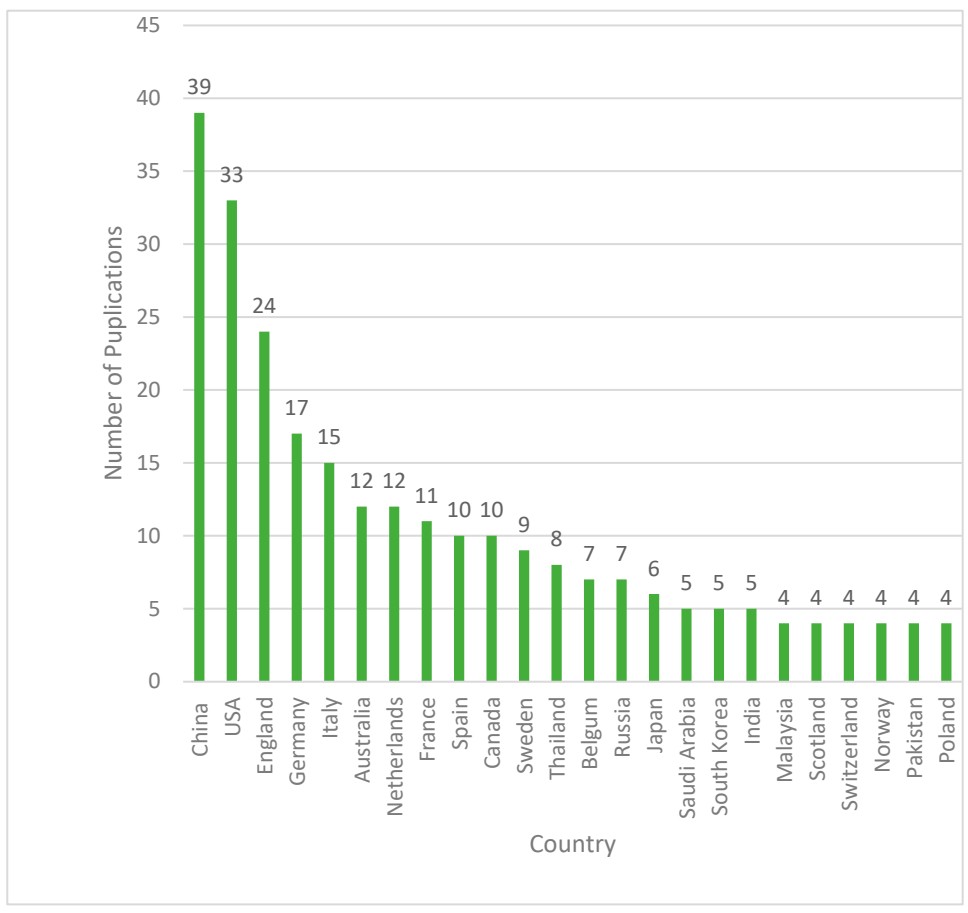

**Figure 3.** Number of publications by country.

## 5. Discussion

This section will discuss the results obtained concerning our research questions and objectives. The discussion also includes future recommendations for energy reporting studies in answer to question four and the conceptualised framework in Figure 4.

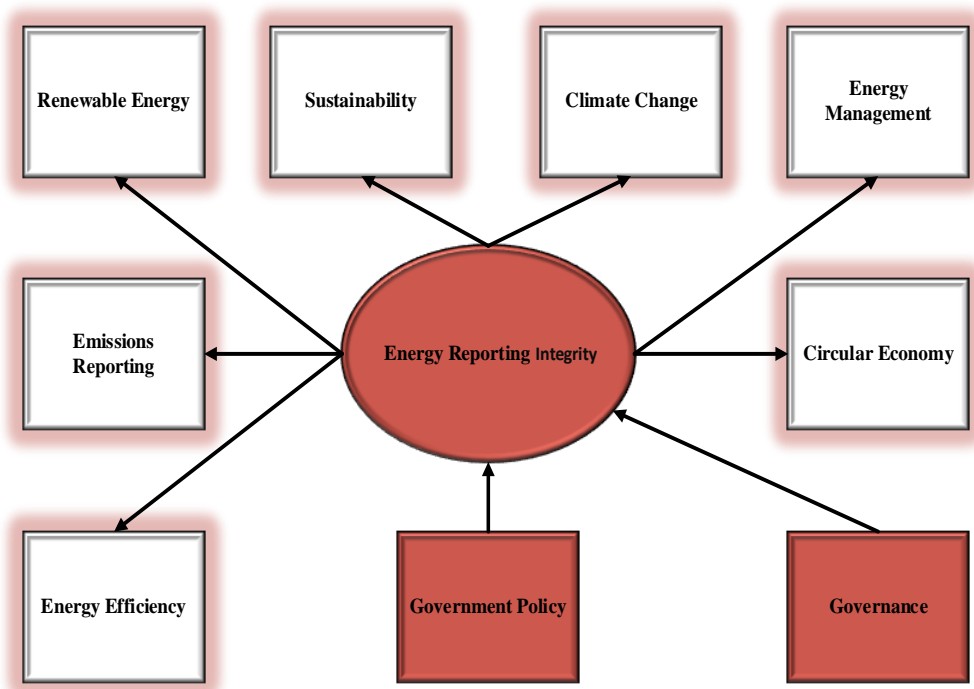

**Figure 4.** A conceptual framework for energy reporting integrity.

*5.1. First Research Question: What Are the Key Themes in Energy Reporting Integrity?*

The key themes of energy reporting integrity are heavily concentrated in studies on energy policy, climate change, energy efficiency, renewable energy, life cycle assessment, carbon emissions, and sustainability. This conclusion is supported by several energy management studies in the literature, including [82,98–100]. However, scholars have investigated various non-financial reporting study areas, frequently yielding incomplete or unclear conclusions. Therefore, the literature gap is confirmed, which we discovered through extensive literature assessment. According to our research, experts have looked mainly at energy reporting in large corporations. There is a scarcity of studies on the integrity of energy reporting in small and medium-sized businesses and non-governmental organisations. According to Rosati and Faria [98], companies that reported sustainable development goals (SDGs) have been in nations with high CSR awareness, fulfilment, individualism, employee rights, power distribution, and long-term awareness. By studying non-financial reports, Landrum and Ohsowski [99] discovered that energy report disclosures were prompted by the gains they offered to the enterprise.

This suggests that sustainability and energy reports were not ingrained in business culture. Corporations frequently utilise non-financial reporting, such as energy consumption reports to access essential sources such as financing and customer relations to 'greenwash' their reputation [100]. Due to a lack of transparency, information on sustainability reports to stakeholders is insufficient [101]. Firms' disclosures usually contain ambiguous explanations for missing key data and limit non-financial declarations with stakeholders [102]. In this context, Boiral et al. [103] investigated stakeholders' opinions of non-financial reporting quality and discovered that global reporting initiatives (GRI) guidelines were irregularly adopted and frequently amended to meet the demands of businesses.

Annual energy reporting only appears in the form of declarative reports within non-financial reports. Many barriers, such as the complexity and volume of data points across numerous intensive-industry operations, make energy reports less critical. Aside from that, there are no clear criteria for selecting and interpreting energy usage data for reporting. Furthermore, this is due to a lack of top-level commitment, energy, expertise, and awareness among the staff [8]. As a result, energy teams' lack of knowledge will not allow them to

convert existing energy consumption data into meaningful reported data. Another difficulty is that manual reporting makes the energy teams' job more difficult. There is a shortage of technology, such as blockchain or advanced metering linked to energy management software. As a result, manual reporting takes longer, fewer reports are issued, and upper management levels monitor the situation infrequently [31].

Governance influences the integrity of energy reporting. The number of board members, the ratio of independent and female directors, the inclusion of an audit committee, and the regularity of board meetings are all factors that can influence energy reports' integrity. A large board typically has more expertise, leading to more significant job scheduling and reliable and transparent non-financial reporting. Because of their predisposition toward acting responsibly and transparently, and sensitivity to societal and environmental issues, such directors are more closely aligned with global reporting initiatives criteria [104]. The same is true for corporate social and audit committees, resulting in improved quality of energy consumption reports [105]. Companies may use non-financial reports to communicate their non-financial performance to enhance stakeholders' opinions. Companies with a practical governance framework can report on what they have accomplished environmentally and socially [106].

### 5.2. Second Research Question: What Is the Role of Government Policy in Energy Reporting Integrity?

The answer to the second research question confirms the research gap regarding government policy and its influence on the integrity of energy consumption reporting. Much of the research found was related to energy management [8], renewable energy [107], energy audits [108], voluntary greenhouse reporting [30], circular economy [109], sustainability issues in production [110], governance [9], and energy policy [111]. Despite the importance of energy policies, such as reporting regulation and incentives programs, scholars have not covered the integrity of energy reporting topics in depth. In the best cases, energy consumption reporting was discussed only as part of sustainability issues [112], not as an independent research area that potentially impacts energy consumption reduction. In a study of internal and external pressures' impact on emissions management techniques and reporting methods, [113] discovered that the critical internal driver is policies and procedures related to emissions or energy reductions within the firm. Besides that, the external driver is the degree of commitment of policymakers and NGOs.

Through an intensive literature review, we found that government policy plays a vital role in shaping the integrity of energy reports. The study results follow Turzo et al.'s [30] findings. Governments and financial regulators are the most active participants in issuing and updating reporting requirements and guidance, followed by stock exchanges and industry bodies. Besides that, companies in the modern era cannot deny investors' requests for various non-financial reports that have an increasingly important role in assessing business continuity [114]. This positive trend is due to the growing stakeholder scrutiny of companies' social and environmental behaviour and the increasing tendency towards legislative obligations to release non-financial reports.

### 5.3. Third Research Question: What Are the Literature's Highest Co-Occurrence Keywords of Energy Reporting?

We get the answer to the third research question based on keyword analysis. The findings revealed useful information regarding the keyword energy report integrity and the distribution of government policies, laying a solid foundation for keyword co-occurrence frequency, as shown in Table 1. The keywords from the government policy and energy report appear in the topics of carbon footprint, climate change, and circular economy. The keyword energy reporting also co-exists in energy efficiency, energy security, energy transition, energy policy, environmental sustainability, governance, renewable energy, resource efficiency, and sustainable development literature. The highest co-occurrence frequency was with the circular economy, sustainability, climate change, and renewable energy topics.

On the other hand, weak co-occurrence frequency was recorded with climate policy, biodiversity, ecosystem services, energy efficiency, energy security, and environmental sustainability topics. The keyword co-occurrence frequency helped to identify which articles or topics discussed energy reports and government policy. Therefore, the topics or articles with weak co-occurrence frequency implied a literature lack or gap with regard to energy reporting and government policy.

*5.4. Fourth Research Question: What Will Be the Future Research of Energy Reporting Integrity Studies?*

The extensive literature review and content analysis allowed the authors to outline future directions for energy consumption disclosure. The conceptualised framework in Figure 4 draws the role of government incentives and governance as enablers for energy report integrity among different sectors. Government intervention through policies and regulations is vital for promoting transparent energy reports. The day-watchman approach supports the result, where governments use policies to reform incentives and penalties for good or bad performers [11]. The regulator (government) informs energy consumers about transparent disclosure, applies incentives or penalties to enforce the law, and acts as the day-watchman [16]. The coordination of policy intervention through incentives enhances the effectiveness of energy use and reporting and helps the government fulfil its obligations for emissions reductions.

Policy interventions and pertinent agency activities promote transparency of energy use disclosures and environmental information reporting through policies and governance measures [115]. The findings by Jacoby et al. [115] support the conceptualised framework where government policy and governance are the external drivers of energy reporting integrity. To demonstrate how well businesses are upholding their environmental duties, there is a growing requirement for them to provide more information about their energy use and environmental information reporting [116]. However, few studies have examined the direct effects of transparency and integrity in energy reporting on sustainable development, emissions reporting, the circular economy, management and efficiency of energy use, adopting renewable energy sources, and climate change. Therefore, the energy report's government policy and integrity are conceptualised into a framework based on critical themes from the literature review, as shown in Table 1 and Figure 4. The conceptualised framework is a guide for future research regarding government policy and the integrity of energy reporting.

Government policy is the driving force that guides the manufacturing and commercial industries towards transparent energy reports. The governance function monitors the integrity of energy reports through auditing. The integrity of energy reports allows the firms to understand their energy efficiency, emissions, sustainability, and climate change impact due to energy consumption. Besides, when transparent energy reports exist, organisations have better measurements of their renewable energy use and energy management effectiveness. According to agency theory, management comprises self-interested individuals who act opportunistically, putting their interests first, even if this is harmful to the owners [117]. As a result, management is monitored by contractual processes such as corporate governance. Given the agency theory perspective, a board of directors and its subcommittees, particularly the audit committee, are required to offer genuine independent oversight of management to ensure that it does not act opportunistically [118]. In the current context, this would focus on verifying the correctness and completeness of energy reporting outputs and analysing management's activities, protecting the company from fines, and safeguarding shareholders by maintaining the integrity of energy reports.

According to the research, businesses respond to external events to enhance their reputation, win the support of regional businesses, the media, and the government, and thus aid in easing public pressure on their operations that have an adverse impact on the environment. In addition, according to Langevin et al. [119], some businesses have a competitive advantage in complying with environmental laws. As a result, their managers may

want to share environmental information to strengthen their relationships with regulators, access more affordable capital, or establish themselves as trustworthy business partners.

Information and motivation problems make resource allocation inefficient. However, when government energy policy is in alignment with energy disclosure, this is more suitable for altering organisations' behaviour toward energy use reporting and energy efficiency [120]. Governance comes from auditing the energy reports submitted by firms or organisations. Therefore, governance aids in verifying the authenticity of energy reports [121]. Another concern is affordability and incentives for energy efficiency initiatives. Due to the risk and uncertainty around the predicted energy savings from renovations, funding sources are more constrained [122]. It is conceivable for resources to be allocated to investments in energy efficiency that fall short of what may be considered a social ideal because of these uncertainties, caused mainly by a lack of timely, accurate, and transparent data. Energy disclosure regulations can boost the quantity and quality of information on energy efficiency and help dispel misconceptions about energy usage, the expected savings from an energy retrofit, and energy conservation strategies [123].

Private investment is significantly impacted by the accuracy of energy reporting, particularly when it comes to renewable energy [124]. Therefore, this study emphasises that research on the integrity of energy disclosures and renewable energy adoption are closely linked. Locating groups of inefficient or energy-intensive businesses or structures allow municipalities and energy suppliers to choose where to focus their capital funding and how to best target energy efficiency incentives [125]. Energy disclosure could also be used to identify potential locations for decentralised power plants and other shared renewable energy capacity types. Transparent energy consumption reports are an indicator of sustainability and an effective way for climate change mitigation. Understanding the geographic trends of energy consumption has the advantage of enabling management of growth prospects and handling the hidden costs of inefficient facilities, such as pollution levels, greenhouse gas emissions, energy management and efficiency, circular economy, and other societal issues [126].

## 6. Managerial Implications

Capital providers can learn whether a company develops long-term value by combining monetary and non-monetary data in the firm's annual disclosure [127]. Non-financial disclosure aids in the development of a firm's corporate credibility. Earnings quality improves when non-financial reports are of high quality [128]. Establishing an audit and a CSR committee enhances reports' transparency and integrity. Managerial attitudes toward non-financial reports, such as energy reports, significantly impact stakeholders' perceptions of non-financial corporate operations, which is critical for a company's legitimacy. Non-financial reports must adhere to the statutory and informal institutional requirements of the country in which it works to be effective. Accurate environmental reporting has a positive impact on brand success and reliability.

Non-financial reporting should provide a comprehensive set of data on a firm's environmental performance, covering carbon emissions, energy consumption, and water consumption [103]. The findings also show that energy disclosure transparency is increased through governance and government policy procedures. Businesses react to external pressure to improve their reputation and get the support of local businesses, the media, and the government, which helps to reduce public pressure on their operations. In addition, compliance with environmental rules gives businesses a competitive edge, and managers frequently divulge environmental information to build rapport with regulators, gain access to more energy sources, or position their companies as reliable business partners.

## 7. Conclusions

Public and private sectors must conduct transparent and integrated energy consumption reporting. The significance is derived from the efforts needed to mitigate climate change and improve sustainability in a long-term approach. Transparent disclosure of en-

ergy consumption illustrates a firm's effectiveness toward sustainability and consumption of scarce resources, which also implies the firm's responsiveness toward environmental issues and government policies on energy efficiency. Furthermore, investors are more willing to join the organisation when information is disclosed, as financial information and non-financial reports are available. In addition, stakeholders and investors will have more confidence in firms that manage their resources efficiently, as reflected in organisation reports. Therefore, energy planning and long-term applications for energy consumption reduction depend on the integrity of energy reporting, which is supported by government policy.

Additionally, the business environment has shifted toward more comprehensive reporting, and the inclusion of non-financial reports with financial reports. Once energy reporting is conducted transparently, the government, stakeholders, and investors can verify a firm's sustainability values. It will also be possible to acknowledge a firm's economic, environmental, social, and governance performance. On the other hand, prior studies have claimed that the quality of CSR reports has been questioned, claiming that corporations are more likely to publish altered reports [106]. In this sense, the credibility of the revealed energy report is improved by the integrity of the energy report [129]. In addition, firms provide open sustainability reports demonstrating their outstanding dedication to sustainability activities [130]. In terms of method, social network analysis is useful for examining the research trend and future direction; however, this analysis requires additional empirical testing to prove its applicability in the actual scenario.

The challenges of energy consumption disclosure can be overcome by concentrating on three–four key areas: first, through government intervention [1] via energy policy and incentives for transparent energy disclosure, such as financial penalties, taxation, and subsidisation; and second, the enactment of specific laws for handling various environmental issues. Thirdly, implementation will meet difficulties even after a comprehensive set of regulations for transparent energy reporting has been passed. When laws are not obeyed, the day-watchman strategy is advantageous. The government may assign sanctions for law violations, and benefits for complying should be provided. Lastly, a global effort should be made to promote sustainable production and consumption while reducing the emissions of $CO_2$.

This extensive literature review suggests that government policy research toward energy consumption reporting is still limited, especially with the absence of a consistent disclosure framework. Therefore, independent research on designing tailored government policies about public and private sector energy reporting is needed. In addition, it is necessary to investigate the contingencies regarding the integrity of energy reporting due to the intensity of energy consumption, the complexity of processes, and the management teams' experience and knowledge level to conduct valuable, relevant energy reports. Future research may examine the benefits of completing the energy report integrity emphasised in this study. Benefits include increased legitimacy from stakeholders, a competitive edge, and gaining an advantage in local and international contractual agreements. Furthermore, regional or country-level studies, in addition to this study, may provide different implications based on the regulatory environments of their contexts.

**Author Contributions:** Conceptualisation, M.H.M.A.-M.; methodology, M.H.M.A.-M.; writing—original draft preparation, M.H.M.A.-M.; writing—review and editing, M.H.M.A.-M., Y.F. and M.-L.T. All authors have read and agreed to the published version of the manuscript.

**Funding:** This research received no external funding.

**Institutional Review Board Statement:** Not applicable.

**Informed Consent Statement:** Not applicable.

**Data Availability Statement:** Not applicable.

**Conflicts of Interest:** The authors declare that there is no conflict of interest.

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
