# Peer review of "Assuring Energy Reporting Integrity: Government Policy’s Past, Present, and Future Roles"

_sustainability, doi:10.3390/su142215405_

Round 1
Reviewer 1 Report
paper deals with a review of energy policies - report and integrity. Study is relevant to the journal. Some comments to improve are:
- improve abstract by stating need for the study
- what are the research questions focussed by the work
- why the questions are iimportant? How the answer is different from previous works?
- there are key reviews in energy domain, authors must consider them and provide key extensions from those studies such as https://doi.org/10.1016/j.rser.2016.12.030G, https://doi.org/10.3390/en14206824
- conclusion can be well elaborated with key knowledge and solid future directions
- review framework must be explained in detail
Author Response
Fifth Reviewer Comments
Paper deals with a review of energy policies - report and integrity. Study is relevant to the journal. Some comments to improve are:
- Improve abstract by stating need for the study.
The abstract has been improved according to the reviewer's valuable comments.
- What are the research questions focused by the work.
Research questions have been listed in the introduction section.
- What are the key themes in energy reporting integrity? This research question will help to clarify both dominant topics and gaps in this research field.
- What is the role of government policy in energy reporting integrity? The response to this question provides an overview and an interpretation of the essential publication in the public policy field, making it easier for researchers interested in government policy studies to understand the latest developments.
- What are the highest co-occurrence keywords of energy reporting in literature? This point makes it possible to understand the most used words in energy reports integrity and its main components that apply to researchers to progress research forward. They must acquire a comprehensive framework.
- What will be the future research of energy reporting integrity studies? The answer to this question can be used to create a research strategy that will aid in developing the topic.
- Why the questions are important? How the answer is different from previous works?
- The first research question will help to clarify both dominant topics and gaps in this research field.
- The response to the second research question provides an overview and an interpretation of the essential publication in the public policy field, making it easier for researchers interested in government policy studies to understand the latest developments.
- The answer to the second research question will make it possible to understand the most used words in energy reports integrity and its main components that apply to researchers to progress research forward. First, they must acquire a comprehensive framework.
- The answer to the fourth research question can be used to create a research strategy that will aid in developing the topic.
- There are key reviews in energy domain, authors must consider them and provide key extensions from those studies such as https://doi.org/10.1016/j.rser.2016.12.030G, https://doi.org/10.3390/en14206824
Key extensions have been added according to the reviewer's valuable advice.
Energy is a vital contributor to all sectors of the economy in general, and especially to industry. Nevertheless, using energy for production typically leads to the creation of undesired emissions that impact the natural ecosystem and the climate globally [1]. Therefore, researchers and government leaders are encouraged to focus on social welfare and energy-related emission assessments and their possible contamination of the environment.
Since conventional energy sources cause ecological problems and significantly contribute to climate change, clean energy is preferable for meeting the public's energy needs [6].
- Conclusion can be well elaborated with key knowledge and solid future directions
The conclusion has been elaborated according to the reviewer valuable comment.
The challenges of energy consumption disclosure can be overcome by concentrating on four key areas: first, through government's intervention [1] through energy policy and incentives for transparent energy disclosure, like financial penalties, taxation, and subsidization; and second, the enactment of specific laws for handling various environmental issues. Thirdly, implementation will meet difficulties even after a comprehensive set of regulations for transparent energy reporting has been passed. When laws are not obeyed, the day-watch-man strategy is advantageous. In addition, the government may assign sanctions for law violations, and benefits for complying should be provided. Lastly, a global effort should be made to promote sustainable production and consumption while reducing the emissions of CO2.
The extensive literature review suggests that government policy research toward energy consumption reporting is still limited, especially with the absence of a consistent disclosure framework. Therefore, independent research on designing tailored government policies about public and private sector energy reporting is needed. In addition, it is necessary to investigate the contingencies regarding the integrity of energy reporting due to the intensity of energy consumption, the complexity of processes, and the management team's experience and knowledge level to conduct valuable, relevant energy reports. Future research may look at the benefits of completing the energy report integrity emphasized in this study. Benefits include increased legitimacy from stakeholders, a competitive edge, and gaining an advantage in local and international contractual agreements. Furthermore, regional or country-level studies, in addition to this study, may provide different implications based on the regulatory environments of the contexts.
- Review framework must be explained in detail.
Framework review has been explained in detail.

Reviewer 2 Report
Dear Editor and Authors
The manuscript explores government policy's past, present, and future role in assuring energy reporting integrity. Although the topic is relevant, I believe that major improvements are needed to make it more useful to the scientific community.
The main reasons for such a recommendation are listed below:
1. Keywords: Should be improved.
2. Highlights: Should be added.
3. Graphical Abstract: Should be added.
4. List of Abbreviations: Should be added.
5. More figures should be added, as well as more explanation in the body of the paper should be added for Figures 1, 2, and 3.
6. Figure 1 should be improved; in its current state, I find it not clear.
7. The introduction section should be improved to better highlight the research performed in this area of research.
8. Methods: The Methods section should be improved.
9. Future recommendations should be added at the end of the conclusions section.
10. References should be improved. Please update all references and use references from the sustainability Journal since only one reference is used from the sustainability Journal.
In conclusion, I believe this manuscript is worthy of publication in the Sustainability Journal, after a major revision.
Author Response
First Reviewer Comments
The manuscript explores government policy's past, present, and future role in assuring energy reporting integrity. Although the topic is relevant, I believe that major improvements are needed to make it more useful to the scientific community.
The main reasons for such a recommendation are listed below:
- Keywords: Should be improved.
Keywords have been improved by adding keywords according to the reviewer precious comments.
- Highlights: Should be added.
Not clear what is meant by the highlight.
- Graphical Abstract: Should be added.
Graphical abstract has been added.
- List of Abbreviations: Should be added.
A list of abbreviations has been added according to the reviewer's valuable recommendation.
- More figures should be added, as well as more explanation in the body of the paper should be added for Figures 1, 2, and 3.
One more figure has been added “Publications number by country”
- Figure 1 should be improved; in its current state, I find it not clear.
Figure one describes the four stages involved, which include 4 stages: The first stage includes choosing Web of Science and Scopus as the main publication's source. The total number of articles were 865 for WoS and 1276 for Scopus. The next step was filtering those journals using the keywords relevant to our study. The retrieved papers were 259. The second stage used Mendeley to filter duplicates. The third stage is a content analysis using VosViewer software for keyword Co-occurrences analysis, themes in literature, and gap identification.
- The introduction section should be improved to better highlight the research performed in this area of research.
The introduction part has been improved. Thanks for the reviewer advice.
- Methods: The Methods section should be improved.
The methods section has been improved.
- Future recommendations should be added at the end of the conclusions section.
The future recommendation has been relocated according to the reviewer's valuable advice.
- References should be improved. Please update all references and use references from the sustainability Journal since only one reference is used from the sustainability journal.
References have been improved, and more references were used from the sustainability journal.

Reviewer 3 Report
The authors should be consider:
1. To cite and describe Figure 1.
2. Check the title of Figure 2 (Figure 2. This is a figure. Schemes follow the same formatting).
Author Response
Second Reviewer Comments
The authors should consider:
- To cite and describe Figure 1.
Figure one is a VISIO drawing of the research procedure.
- Check the title of Figure 2 (Figure 2. This is a figure. Schemes follow the same formatting).
Figure 2 title has been edited accordingly. Thanks for the reviewer valuable comment.

Reviewer 4 Report
Authors should read the following comments and recommendations:
(1) Lack of listed main objective of the study in the abstract.
(2) Numerous editorial errors (double spaces, missing sources under figures and their incorrect titling e.g. Figure 2).
(3) Many linguistic errors.
(4) The section with limitations and future research directions could be combined with the discussion section.
(5) The sections with method description and results have the same designation (3).
(6) To be expanded the section with research results. The authors should do a more detailed quantitative analysis. E.g. show the distribution of publications by year, the structure of publications by journal or field of study.
(7) In the discussion, the authors should also refer to the limitations of the adopted research procedure.
(8) In conclusion, the authors should refer to the main objective and research questions.
The main weaknesses of the study are editorial and linguistic shortcomings that make the article difficult to read.
Author Response
Third Reviewer Comments
(1) Lack of listed main objective of the study in the abstract.
The objective of the study has been added according to reviewer’s valuable comment.
(2) Numerous editorial errors (double spaces, missing sources under figures and their incorrect titling e.g., Figure 2).
Editing of the paper has been conducted. The title of Figure 2 has been edited.
(3) Many linguistic errors.
Proofreading has been conducted accordingly.
(4) The section with limitations and future research directions could be combined with the discussion section.
The limitation and future research section have been combined with the discussion section according to the reviewer's valuable comments.
(5) The sections with method description and results have the same designation (3).
The designation has been edited accordingly. We appreciate the reviewer's comment.
(6) To be expanded the section with research results. The authors should do a more detailed quantitative analysis. E.g., show the distribution of publications by year, and the structure of publications by journal or field of study.
Research results have been expanded according to the author's precious comments.
(7) In the discussion, the authors should also refer to the limitations of the adopted research procedure.
In terms of method, social network analysis is useful for examining the research trend and future direction; however, this analysis requires additional empirical testing to prove its applicability in the actual scenario.
(8) In conclusion, the authors should refer to the main objective and research questions.
The reviewer's concern has been addressed. Finally, we discussed the main objective and research questions to discuss the conclusion.
The study's main weaknesses are editorial and linguistic shortcomings that make the article difficult to read.
We have improved the editorial and linguistic issues accordingly.

Reviewer 5 Report
This work is very intriguing; it's well-written and grounded in research. However, I think there is a room for some improvements to raise its quality.
1) The authors write about "policy's success" and discuss how "to enhance energy policy effectiveness". I feel that this paper could refer to some framework enabling qualification on the policy success/failure. See, e.g. doi.org/10.1016/j.enpol.2021.112745
This would also allow for the establishment of a connection to a global framework based on the SDGs and the 2015 Paris Agreement.
2) The authors analyse "Government interventions through energy policy", write how this guides and shapes "the integrity of energy consumption reporting and energy management practices", as well as discuss how "the government enforces reporting transparency through energy policy and encourages energy users to cooperate closely", and how "government regulation drives the industry to carry out its social and environmental responsibilities".
I believe that a theory of public regulation would help to ground this analysis and make it more thorough. Look at this example to see how it was done doi.org/10.1016/j.enpol.2022.112914 (discussion on the day-watchman approach).
Author Response
Fourth Reviewer Comments
This work is very intriguing; it's well-written and grounded in research. However, I think there is a room for some improvements to raise its quality.
1) The authors write about "policy's success" and discuss how "to enhance energy policy effectiveness". I feel that this paper could refer to some framework enabling qualification on the policy success/failure. See, e.g. doi.org/10.1016/j.enpol.2021.112745
This would also allow for the establishment of a connection to a global framework based on the SDGs and the 2015 Paris Agreement.
Governmental subsidies for conventional fuels may reduce energy efficiency [10]. If there is no suitable policy in place, energy subsidies may result in higher energy use and carbon dioxide emissions. Elgouacem [11] argues that energy policy is unsuccessful when subsidies lead to increased energy use and carbon dioxide emissions as a result of low energy costs. As a result, the transition to renewable energy will be delayed. Moerenhout et al. [12] discovered that energy policy support in the form of subsidies reduces competition for renewable energy sources. Rentschler et al. [13] stated that energy subsidies disincentivize investment in alternative energy, impede innovation and energy efficiency, and increase fiscal responsibilities. Reforming the subsidies will, however, raise the production and consumption costs of externalities related to fossil fuels. The right distribution of incentives for energy management and reporting is, therefore, necessary for the effectiveness of energy policy. For the transition to alternative fuels and energy efficiency, energy policy must be aligned with incentives to achieve transparent energy reporting targets.
Previous studies have employed the day-watchman technique [14] to strike a balance between the public interest and regulations regarding energy policy. The day-watchman strategy serves as a compromise, balancing the state's and the market's obligations in preserving the public interest. The day-watchman approach includes defined goals such as the ability to create regulations and standards, the ability to provide authorizations and permissions, the legality of public regulation, monitoring and surveillance, and mitigating and penalizing [15]. This strategy creates a framework for regulatory actions in which the regulator (day-watchman) creates the game's rules, informs market participants, and enacts incentives or penalties to enforce the rules [16].
2) The authors analyses "Government interventions through energy policy", write how this guides and shapes "the integrity of energy consumption reporting and energy management practices", as well as discuss how "the government enforces reporting transparency through energy policy and encourages energy users to cooperate closely", and how "government regulation drives the industry to carry out its social and environmental responsibilities".
I believe that a theory of public regulation would help to ground this analysis and make it more thorough. Look at this example to see how it was done doi.org/10.1016/j.enpol.2022.112914 (discussion on the day-watchman approach).
The day-watchman strategy aids in addressing policy deficiencies while safeguarding society and the general public's interests [10]. The degree to which state institutions act, explicitly or implicitly, in addressing energy concerns, such as transparent energy use disclosure through legislation, can be used to examine the challenge for policy intervention through subsidy and government incentive reform [80]. For instance, the government's yearly budget must be approved by the Parliament each year, in which incentives or subsidies are essential components. The day-watchman strategy can be helpful in this situation. Better policies are developed by the regulator (government), who provides information to the energy users about transparent disclosure and uses punishments to enforce the law [16]. The alignment of policy intervention through incentives helps to improve energy use and reporting efficacy and assists the government in meeting its obligations toward emissions reductions.

Round 2
Reviewer 2 Report
I think it can be accepted in its present form.
Author Response
Thank you
Reviewer 4 Report
The authors have incorporated most of the comments identified by the reviewer. The following changes are still recommended: (1) listing the sources under all tables and figures (whether they are the authors' own elaboration or borrowed from others' papers); (2) improving the readability of Figure 3. (including labels and changing the background to white); (3) making the entire reference list compatible with the requirements of the journal.
Author Response
Cover Letter
Dear Sir/Madam
Warmest greetings to you and all sustainability journal editors and reviewers.
We appreciate your kind comments on our review paper titled “Assuring Energy Reporting Integrity: Government Policy's Past, Present, And Future Roles”. Your comments have helped to raise the quality of the paper. Our endless gratitude for the guidance of the reviewers.
Please find the revisions of the manuscript in the next page.
Truly yours,
Authors
Reviewer 4- 2nd Round
- listing the sources under all tables and figures (whether they are the authors' own elaboration or borrowed from others' papers);
- All tables and figures are authors own elaborations. Thanks for reviewer valuable comment.
- improving the readability of Figure 3. (including labels and changing the background to white);
- Figure 3 has been changed according to reviewer precious advice.
- making the entire reference list compatible with the journal's requirements.
- References have been edited according to journal’s guidance
- E,g
- Mardani, A.; Zavadskas, E.K.; Streimikiene, D.; Jusoh, A.; Khoshnoudi, M. A Comprehensive Review of Data Envelopment Analysis (DEA) Approach in Energy Efficiency. Renewable and Sustainable Energy Reviews 2017, 70, 1298–1322, doi:https://doi.org/10.1016/j.rser.2016.12.030.
